# On a $\mathbb{Z}_3$-valued discrete topological term in

# 10d heterotic string theories

Yuji Tachikawa and Hao Y. Zhang

Kavli Institute for the Physics and Mathematics of the Universe (WPI),
University of Tokyo, Kashiwa, Chiba 277-8583, Japan

We show that the low-energy effective actions of two ten-dimensional supersymmetric heterotic strings are different by a $\mathbb{Z}_3$-valued discrete topological term even after we turn off the $E_8 \times E_8$ and $Spin(32)/\mathbb{Z}_2$ gauge fields. This will be demonstrated by considering the inflow of normal bundle anomaly to the respective NS5-branes from the bulk. We also find that the $Spin(16) \times Spin(16)$ non-tachyonic non-supersymmetric heterotic string has the same non-zero $\mathbb{Z}_3$-valued discrete topological term. We will also explain the relation of our findings to the theory of topological modular forms.

The paper is written as a string theory paper, except for an appendix translating the content in mathematical terms. We will explain there that our finding identifies a representative of the $\mathbb{Z}/3$-torsion element of $\pi_{-32}\mathrm{TMF}$ as a particular self-dual vertex operator superalgebra of $c = 16$ and how we utilize string duality to arrive at this statement.

# 1   Introduction and summary

$\mathbb{Z}_3$-**valued discrete topological term:**     There are two supersymmetric heterotic string theories in ten dimensions [GHMR85a, GHMR85b, GHMR86]. Although they have different gauge groups, namely $E_8 \times E_8$ and $Spin(32)/\mathbb{Z}_2$,[1] the field contents of the gravitational sector are common to both and consist of the metric, the dilaton, the $B$-field and their superpartners. Their two-derivative low-energy effective actions of the gravitational sector are also the same, since they are uniquely fixed by supersymmetry. Discrete topological terms in the actions are not constrained in the same way, however, and thus can differ between two supersymmetric heterotic theories. Our main aim in this paper is to show that they are indeed different, by a certain $\mathbb{Z}_3$-valued discrete topological term.

---

[1]We will not be perfectly consistent in our notations concerning the global form of gauge and other groups. For example, $E_8 \times E_8$ should more precisely be $(E_8 \times E_8) \rtimes \mathbb{Z}_2$. We will later refer to the gauge group of the non-tachyonic non-supersymmetric heterotic string theory in ten dimensions as $Spin(16) \times Spin(16)$; the precise global structure in this case does not seem to have been completely written down in the literature, to the knowledge of the authors, although see [BDDM23]. We will be careful at least to make sure that the global form we present does consistently act on the fields in the theory concerned.

To describe the topological term involved, let us first recall that the gravitational sector satisfies the relation

$$dH = \frac{1}{2}p_1, \qquad (1.1)$$

where $H$ is the gauge-invariant field strength of the $B$-field and $p_1 \propto \operatorname{tr} R^2$ is the first Pontryagin class of the spacetime curvature [GS84]. As written above, the equation is at the level of differential forms, but the relation has by now been convincingly shown to be satisfied at a more refined topological level, defining what mathematicians call the *string structure*, see e.g. [Wit85, SSS09, Yon22, BDDM23].

In general, given a spacetime structure $S$, the possible discrete topological terms in $d$ dimensions are known to be classified in terms of the torsion part of the bordism group $\Omega_d^S$ of $d$-dimensional $S$-structured manifolds [FM04, FH16, Yon18]. To be more explicit, we say two $d$-dimensional $S$-structured closed manifolds $M_{1,2}$ are bordant, $M_1 \sim_S M_2$, when there is a $(d+1)$-dimensional $S$-structured manifold $N$ such that its boundary is $\partial N = M_1 \sqcup \overline{M_2}$, where $\overline{M_2}$ is $M_2$ with orientation and other associated geometric structures appropriately reversed. Then the classes $[M]$ of $S$-structured manifolds under the equivalence relation $\sim_S$ form the $S$-bordism group. Now, suppose $[M] \neq 0$ but $n[M] = 0$. Then we can introduce a $\mathbb{Z}_n$-valued topological term detecting the class $[M]$, which assigns an exponentiated Euclidean action $e^{2\pi i k/n}$ to the bordism class $k[M]$.

In the case of the string structure, the computation of the bordism groups was done in [Gia71], with the results given as follows:

| $d$ | 0 | 1 | 2 | 3 | 4 | 5 | 6 | 7 | 8 | 9 | 10 | 11 | 12 | 13 |
|---|---|---|---|---|---|---|---|---|---|---|---|---|---|---|
| $\Omega_d^{\text{string}}$ | $\mathbb{Z}$ | $\mathbb{Z}_2$ | $\mathbb{Z}_2$ | $\mathbb{Z}_{24}$ | 0 | 0 | $\mathbb{Z}_2$ | 0 | $\mathbb{Z}\oplus\mathbb{Z}_2$ | $(\mathbb{Z}_2)^2$ | $\mathbb{Z}_6$ | 0 | $\mathbb{Z}$ | $\mathbb{Z}_3$ |

$$(1.2)$$

This means that a $\mathbb{Z}_6$-valued topological term is possible for ten-dimensional heterotic string theories. We now write $\Omega_{10}^{\text{string}} = \mathbb{Z}_6 = \mathbb{Z}_2 \times \mathbb{Z}_3$. The $\mathbb{Z}_3$ part is known to be generated by the group manifold of $Sp(2)$ with a unit $H$ flux specified by $1 \in H^3(Sp(2), \mathbb{Z}) = \mathbb{Z}$, see e.g. [Hop02, the end of Sec. 2]. Then we can consider a $\mathbb{Z}_3$-valued topological term detecting this class of manifolds. We will show that the gravitational effective actions of two supersymmetric heterotic strings in ten dimensions are different[2] by this $\mathbb{Z}_3$-valued topological term.

**Discrete topological term and the global anomaly of NS5-branes:** The methods we employ are the following. We will start from an analogy. Let us say that a ten-dimensional theory has a Green-Schwarz coupling

$$\int_{M_{10}} B \wedge Y = \int_{M_{10}} H \wedge CS[Y], \qquad (1.3)$$

---

[2]It does *not* make sense to say that two supersymmetric heterotic string theories have theta angles $\theta^{E_8 \times E_8}$, $\theta^{Spin(32)/\mathbb{Z}_2} \in \mathbb{Z}_3$ so that $\theta^{E_8 \times E_8} - \theta^{Spin(32)/\mathbb{Z}_2} = \pm 1 \in \mathbb{Z}_3$. This is because the topological parts of the effective action, $S_{\text{top}}^{E_8 \times E_8}[g, B]$ and $S_{\text{top}}^{Spin(32)/\mathbb{Z}_2}[g, B]$ depend continuously on the metric $g$ and the $B$-field, much as a Chern-Simons interaction does on the gauge field $A$. The difference $S_{\text{top}}^{E_8 \times E_8}[g, B] - S_{\text{top}}^{Spin(32)/\mathbb{Z}_2}[g, B]$ is, however, independent of continuous variations of $g$ and $B$, and it is this difference that equals the non-trivial $\mathbb{Z}_3$-valued topological term detecting $Sp(2)$ with a unit $H$ flux.

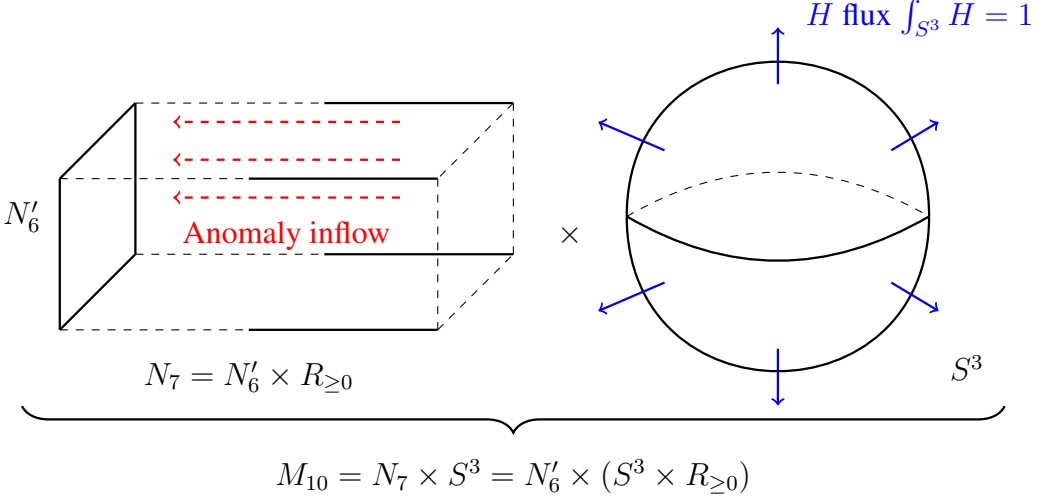

Figure 1: The anomaly inflow from the ten dimensional bulk to the NS5-branes.

where $Y$ is a gauge-invariant 8-form constructed from the gauge and spacetime curvatures, and $CS[Y]$ is the corresponding Chern-Simons term. Now we take $M_{10}$ to be a fibration of the form

$$S^3 \to M_{10} \to N_7, \tag{1.4}$$

with $\int_{S^3} H = k$. Then the integration of (1.3) along the $S^3$ direction gives the coupling

$$k \int_{N_7} CS[Y]. \tag{1.5}$$

This provides the anomaly inflow to the six-dimensional theory on the worldvolume of the stack of $k$ NS5-branes. To see this, we further assume that $N_7$ has the form $N_7 = \mathbb{R}_{>0} \times N_6'$, where the $\mathbb{R}_{>0}$ part has the coordinate $r > 0$, and that the $S^3$ fiber shrinks to zero side at $r = 0$. Now, the $S^3$ fiber and the radial coordinate $r$ form the $\mathbb{R}^4$ normal directions to the stack of the NS5-branes, whose charge is measured by the flux $\int_{S^3} H = k$. Then the worldvolume $N_6'$ of the NS5-branes is on the boundary of $N_7$. Note that we consider a general fibration (1.4) for $M_{10}$, not just a direct product $S^3 \times N_7$. This allows us to capture the anomaly inflow for the rotational symmetry of the normal bundle of the stack of NS5-branes. See Fig. 1 for the schematic description of the mechanism. [3]

In our case, the term of our interest is the $\mathbb{Z}_3$-valued discrete topological term discussed above, detecting the $Sp(2)$ group manifold with an $H$ flux. This term cannot be written as a traditional

---

[3]In general, see [CDLZ20, LY22, BDDM23] for some recent studies of discrete anomalies in string theory and supergravity theories which make use of instantonic objects.

Green-Schwarz term as in (1.3), but is a *global* counterpart to it, in the following sense. Consider the subgroup $Sp(1) \subset Sp(2)$ obtained by sending $q \in Sp(1)$ to $\mathrm{diag}(q,1) \in Sp(2)$, where $q$ is a unit quaternion. With this we can put the $Sp(2)$ group manifold into a fibration of the form (1.4):

$$S^3 = Sp(1) \to Sp(2) \to Sp(2)/Sp(1) = S^7, \tag{1.6}$$

with $k$ units of the $H$ flux through the $S^3$ fiber. Any $Sp(1) = SU(2)$ bundle over $S^7$ is classified by $\pi_6(SU(2)) = \mathbb{Z}_{12}$, and this particular fibration is known to provide its generator [MT64a, MT64b].[4] Therefore, from the seven-dimensional point of view, the phase associated to this fibration (1.6) measures the global anomaly for the normal bundle rotational symmetry of a stack of $k$ NS5-branes. Now, our $\mathbb{Z}_3$-valued discrete topological term in ten dimensions assigns the Euclidean action $e^{2\pi i k/3}$ to this fibration. Therefore, this topological term contributes to the global anomaly of the normal bundle rotational symmetry of a stack of $k$ NS5-branes in a prescribed way.

Luckily, the worldvolume theories on $k$ NS5-branes of both $E_8 \times E_8$ and $Spin(32)/\mathbb{Z}_2$ heterotic string theories are sufficiently well understood, to the extent that we can use our knowledge on them to determine the difference in the global anomalies of the normal bundle rotational symmetry. This allows us to find the difference of the $\mathbb{Z}_3$-valued topological terms of two supersymmetric heterotic string theories in ten dimensions, which turns out to be nonzero.

**NS5-branes as gauge instantons:**  So far we only considered fields in the gravitational sector. As the $Sp(2)$ group manifold with the $H$ flux gives a nontrivial bordism class, there is no $N_{11}$ such that $\partial N_{11} = Sp(2)$, as long as we only consider the metric and the $B$-field on $N_{11}$.

Heterotic string theories, however, have non-Abelian gauge fields as well, and NS5-branes can be regarded as a zero-size limit of their instanton configurations. This means that we can actually find $N_{11}$ such that $\partial N_{11} = Sp(2)$ if we allow a suitable nonzero gauge field $F$ on $N_{11}$, this time solving

$$dH = \frac{p_1}{2} - n \tag{1.7}$$

where $n \propto \mathrm{tr}\, F^2$ is the instanton number density. Indeed, $S^3$ with a unit $H$ flux itself is a boundary of a four-dimensional disk $D^4$ with a one-instanton configuration inside. Then fibering this $D^4$ with the gauge field and the $H$-filed over $S^7$ gives the required $N_{11}$.

As was discussed e.g. in [TY23], there is a method to compute the Euclidean action associated to the $Sp(2)$ manifold with the $H$ flux using such an $N_{11}$. Furthermore, the only essential data

---

[4]Although it is not relevant to the content of this paper, the following fact about this fibration might be of some interest to the readers. Thom famously asked if any integral homology class can be realized by embedded submanifolds. This fibration is known to give a lowest-dimensional counterexample [BHK00]. Namely, the integral homology group of $Sp(2)$ is $\mathbb{Z}$ at degree 3 and 7, but there is no embedded submanifold of dimension 7 realizing the generator of $H_7(Sp(2), \mathbb{Z}) = \mathbb{Z}$. Only three times the generator is thus realized. We should note that there does exist an immersion $f : M_7 \to Sp(2)$ such that $[f(M_7)] \in H_7(Sp(2), \mathbb{Z})$ is the generator. A different example $S^7/\mathbb{Z}_3 \times S^3/\mathbb{Z}_3$, where there is not even an immersion realizing a degree-7 homology class, was given in the first arXiv version of the paper [BHK00]. In both cases the proof is a standard application of basic tools of algebraic topology, and might again be of some interest to the readers.

needed for this computation are the ordinary Green-Schwarz couplings $B \wedge X^{E_8 \times E_8}$ and $B \wedge X^{Spin(32)/\mathbb{Z}_2}$.

This alternative method of computation can be used to determine the $\mathbb{Z}_3$-valued discrete topological term of the non-supersymmetric but non-tachyonic heterotic string theory with gauge group $Spin(16) \times Spin(16)$, first found in [DH86, SW86, AGGMV86, KLT86]. This follows from the fact that the Green-Schwarz coupling $B \wedge X^{Spin(16) \times Spin(16)}$ of the non-tachyonic $Spin(16) \times Spin(16)$ theory satisfies

$$X^{Spin(16) \times Spin(16)} = X^{Spin(32)/\mathbb{Z}_2} - X^{E_8 \times E_8} \tag{1.8}$$

when we restrict the gauge groups on the right hand side to be on the common $Spin(16) \times Spin(16)$ part.[5] We will see that this leads to the conclusion that the $\mathbb{Z}_3$-valued discrete topological term of the $Spin(16) \times Spin(16)$ theory is nonzero.

**Reconstructing 10D Topological Terms**  It is good time to take a pause an go over what we have done at this point. We take the anomaly of the worldvolume theory on instantonic NS5 branes in $E_8 \times E_8$ and $Spin(32)/\mathbb{Z}_2$ heterotic string theories as input, take their difference to get the global anomaly on the "formal difference" NS5 brane in the non-supersymmetric $Spin(16) \times Spin(16)$ heterotic string theory. From the global anomaly of this formal difference, we then take a major leap and reinterpret the anomaly theory associated a 6D instanton worldvolume (for which at least an $SU(2) \subset G$ gauge group is essential) as the 10D discrete theta angle that can only detected by a specific $X_{10}$. In order to go up in dimensionality in this way, we need this $X_{10}$ to be an $S^3$ fibration, whose "filling" $N_{11}$ instantonic configuration of the gauge bundle in the fiber $\mathbb{R}^4$ directions. In the end, we use $SU(2) \cong S^3$ to reinterpret the gauge bundle as extra spacetime dimensions.

Such a reconstruction is reminiscent of a somewhat reverse process [ABGE+21], that is to reduce the topological terms in the string theory spacetime in $M \times X_\Gamma$ (for $X_\Gamma$ an internal geometry with conifold singularity) to obtain the Symmetry Topological Field Theory (SymTFT) for the geometrically engineered QFT in $M$. However, there are two differences that we want to point out: (a) Our analysis is concerned the anomaly theory $\mathcal{A}$, which admits the original QFT as its only boundary. On the other hand, symmetry TFT admits a pair of boundaries - a dynamical boundary and a topological boundary.[6] (b) The dimension reduction approach of generating symmetry TFT from geometric engineering considers $M \times X_\Gamma$ as a *direct product*. For our reconstruction, we do not need to explicitly write down the reconstructed 10D topological action, but we only need that action to be detectable by the special spacetime configuration that is a *non-trivial fibration*.

We mention in passing that an exotic IIB topological action has been inferred in [DDHM21] via global anomaly cancellation associated to the $SL(2, \mathbb{Z})$ bundle (and its suitable finite cover).

---

[5] $Spin(16) \times Spin(16)$ is not quite a subgroup of neither $Spin(32)/\mathbb{Z}_2$ nor $E_8 \times E_8$, but there do exist homomorphisms $Spin(16) \times Spin(16) \to Spin(32)/\mathbb{Z}_2$ and $Spin(16) \times Spin(16) \to E_8 \times E_8$, which give finite covers of the respective images. This suffices for our purposes.

[6] For more mathematically-oriented readers, the symTFT is formulated in the "quiche construction" as in [FMT22].

It would be interesting to make connection between these topological terms obtained in different string duality frames.

**Relation to topological modular forms:** Our findings also have bearings on the mathematical theory of topological modular forms (TMFs) in the context of the proposal of Stolz and Teichner [ST04, ST11], which says that TMF classes are realized by two-dimensional $\mathcal{N}=(0,1)$ supersymmetric theories. It would suffice to say here the following. Any $c_L = 16$ internal worldsheet theory $T$ for a ten-dimensional heterotic string theory determines an element $[T] \in \mathrm{TMF}_{-32}$, the Abelian group of topological modular forms of degree $-32$. A formal integer linear combination $\sum_i n_i T_i$ similarly gives an element $\sum_i n_i [T_i] \in \mathrm{TMF}_{-32}$. An integer linear combination of theories $\sum_i n_i T_i$ whose total elliptic genus vanishes gives an element in an important subgroup $A_{-32} \subset \mathrm{TMF}_{-32}$, which is known to be $\mathbb{Z}_3$. Then our main results can be summarized by saying that

$$[T^{Spin(16)\times Spin(16)}] = [T^{Spin(32)/\mathbb{Z}_2}] - [T^{E_8 \times E_8}] \in A_{-32} \tag{1.9}$$

is a nontrivial generator of this group $A_{-32} = \mathbb{Z}_3$, where we used $T^G$ for the internal worldsheet theory of the heterotic string theory with gauge group $G$.

**Structure of the paper:** The rest of the paper is organized as follows. In Sec. 2, we start by analyzing the anomaly inflow to NS5-branes in both $Spin(32)/\mathbb{Z}_2$ and $E_8 \times E_8$ heterotic string theories. This is mostly a review of known materials.

In Sec. 3, we use the data gathered in Sec. 2 to evaluate the inflow of the global anomaly of the normal bundle rotational symmetry to NS5-branes, leading to our first main conclusion as explained above, that the gravitational couplings of the two ten-dimensional supersymmetric heterotic string theories are different by a $\mathbb{Z}_3$-valued discrete topological term. The computational techniques are known but not as widely as the content of Sec. 2. As such we will detail the steps of the computation.

In Sec. 4, we explain how the same result can be arrived, at least in principle, by a computation which uses NS5-branes which are not point-like but are realized as an instanton configuration of the non-Abelian gauge fields of the heterotic string theories in question. This will allow us to come to the second main conclusion of ours, namely that the non-tachyonic $Spin(16) \times Spin(16)$ heterotic string theory also has this nonzero $\mathbb{Z}_3$ topological term.

In Sec. 5, we explain what our results mean in the context of topological modular forms and the proposal of Stolz and Teichner. This will be done by connecting to the results of [TY23].

We have a single Appendix, Appendix A, in which we explain our findings in a language hopefully more palatable to mathematicians. We will carefully distinguish which part of our arguments can be made into rigorous mathematics and which part uses string duality not readily translatable into mathematics yet.

# 2 Anomaly inflow to NS5-branes

Here we study the perturbative anomaly inflow to a stack of $k$ NS5-branes in two supersymmetric heterotic string theories. We use the convention that $Q$, $T$ and $N$ denote the tangent bundle to the ten-dimensional spacetime, the tangent bundle to the six-dimensional worldvolume of the NS5-brane, and the normal bundle to the NS5-brane, respectively. As vector bundles, we have $Q = T \oplus N$. The $Spin(4)$ rotational symmetry of $Q$ is decomposed into $SU(2)_L \times SU(2)_R$, where $SU(2)_R$ is the R-symmetry of the $\mathcal{N}=(0,1)$ supersymmetry preserved by the NS5-branes.

Our normalization of the Pontryagin classes are the standard ones, where we have $p_1(F) = -\operatorname{tr} F^2/2$ and $p_2(F) = (\operatorname{tr} F^2/2)^2/2 - \operatorname{tr} F^4/4$ for $F$ being an $so$-valued curvature 2-form with $(2\pi)^{-1}$ included. We then have $p_1(N)/2 = -c_2(L) - c_2(R)$ and $\chi(N) = c_2(L) - c_2(R)$, where $c_2(L)$ and $c_2(R)$ are the second Chern classes of $SU(2)_L$ and $SU(2)_R$, respectively.

## 2.1 Anomaly polynomials in ten dimensions

Let us first record the ten-dimensional fermion anomaly polynomials $I_{12}$ of the two supersymmetric heterotic theories, which have factorized forms. For the $Spin(32)/\mathbb{Z}_2$ theory, we have

$$I_{12}^{Spin(32)/\mathbb{Z}_2} = X_4^{Spin(32)/\mathbb{Z}_2} Y_8^{Spin(32)/\mathbb{Z}_2} \tag{2.1}$$

with

$$X_4^{Spin(32)/\mathbb{Z}_2} = \frac{p_1(Q)}{2} - \frac{p_1(F)}{2}, \tag{2.2}$$

$$Y_8^{Spin(32)/\mathbb{Z}_2} = \frac{3p_1(Q)^2 - 4p_2(Q)}{192} - \frac{1}{12}\frac{p_1(Q)}{2}\frac{p_1(F)}{2} + \frac{p_1(F)^2 - 2p_2(F)}{12}, \tag{2.3}$$

where $F$ stands for the $Spin(32)/\mathbb{Z}_2$ gauge bundle. For the $E_8 \times E_8$ theory, we have

$$I_{12}^{E_8 \times E_8} = X_4^{E_8 \times E_8} Y_8^{E_8 \times E_8} \tag{2.4}$$

with

$$X_4^{E_8 \times E_8} = \frac{p_1(Q)}{2} - n - n', \tag{2.5}$$

$$Y_8^{E_8 \times E_8} = \frac{3p_1(Q)^2 - 4p_2(Q)}{192} - \frac{1}{12}\frac{p_1(Q)}{2}(n + n') + \frac{n^2 - nn' + n'^2}{6}, \tag{2.6}$$

where $n$, $n'$ are instanton numbers of two $E_8$ gauge bundles.

These anomalies are canceled by the Green-Schwarz mechanism: we introduce a $B$-field whose gauge-invariant field strength $H$ satisfies

$$dH = X_4, \tag{2.7}$$

and we further require the Green-Schwarz interaction term

$$-\int_{10d} B \wedge Y_8. \tag{2.8}$$

This produces the cancelling anomaly $-X_4 Y_8$, which works for both $E_8 \times E_8$ and $Spin(32)/\mathbb{Z}_2$ heterotic string theories.

## 2.2   NS5-branes in the $Spin(32)/\mathbb{Z}_2$ heterotic string

NS5-branes in the $Spin(32)/\mathbb{Z}_2$ heterotic string theory are D5-branes in the Type I string theory. As such, the worldvolume spectrum can be straightforwardly found by quantizing open strings. The result is an $\mathcal{N}=(0,1)$ supersymmetric gauge theory with $Sp(k)$ gauge algebra, with a hypermultiplet in the antisymmetric of $Sp(k)$, together with a half-hypermultiplet in the bifundamental representation of $Sp(k) \times SO(32)$. As for the normal bundle symmetry, the gauginos are doublets of $SU(2)_R$, the hyperinos in the antisymmetric are doublets of $SU(2)_L$, and the hyperinos in the bifundamental are neutral. These facts were first found in [Wit95, Sch95].

This information is enough to compute the anomaly polynomial $I_8^{Sp(k)}$ of the six-dimensional worldvolume theory. We find that

$$I_8^{Sp(k)} - kY_8^{Spin(32)/\mathbb{Z}_2} = Z_4^{Spin(32)/\mathbb{Z}_2} W_4^{Spin(32)/\mathbb{Z}_2} \tag{2.9}$$

where

$$Z_4^{Spin(32)/\mathbb{Z}_2} = X_4^{Spin(32)/\mathbb{Z}_2} + k\chi(N), \qquad W_4^{Spin(32)/\mathbb{Z}_2} = k\frac{p_1(T) - p_1(N)}{24} - q_1(G) \tag{2.10}$$

where $G$ is the $Sp(k)$ gauge field and $q_1(G) = -\operatorname{tr} G^2/2$ is its instanton number.

The $H$-field now satisfies

$$dH = Z_4^{Spin(32)/\mathbb{Z}_2} = \frac{p_1(Q)}{2} - \frac{p_1(F)}{2} + k\chi(N), \tag{2.11}$$

Here, the additional contribution $k\chi(N)$ comes from the fact that an $H$ field through $S^3$ fiber around the stack of NS5-branes, with $\int_{S^3} H_{\text{fiber}} = k$, leads to $dH_{\text{fiber}} = -k\chi(N) = -k(c_2(L) - c_2(R))$. Then writing $H_{\text{total}} = H_{\text{fiber}} + H_{\text{base}}$ yields (2.11), where $H$ in the equation stands for $H_{\text{base}}$.[7] We also have an additional Green-Schwarz coupling on the NS5-brane worldvolume, given by

$$-\int_{\text{6d}} B \wedge W_4^{Spin(32)/\mathbb{Z}_2}. \tag{2.12}$$

In the type I frame, this is a specialization of the general D-brane coupling that is proportional to

$$\int \left(\sum C_p\right) \wedge \sqrt{\hat{A}(T)/\hat{A}(N)} \operatorname{tr} e^{iF}. \tag{2.13}$$

The combination of (2.11) and (2.12) produces the anomaly which cancels the difference of the fermion anomaly $I_8^{Spin(32)/\mathbb{Z}_2}$ and the inflow term $kY_8^{Spin(32)/\mathbb{Z}_2}$ given in (2.9). This confirmation of the anomaly inflow mechanism including the normal bundle contributions on the stack of $k$ NS5-branes of the $Spin(32)/\mathbb{Z}_2$ theory was first done in [Mou97], and we simply reproduced it in our notations.[8]

---

[7]This relation $dH \propto c_2(L) - c_2(R)$ is the reason how the WZW term reproduces the fermion anomaly in the non-Abelian bosonization [Wit84, Wit83]. The same relation was also pointed out in the context of the normal bundle anomaly cancellation in [Wit96].

[8]See also the paper [IMY10] where the anomaly cancellation was studied from the point of view of the NS5-brane realized as an instanton.

We pause here to mention that the fermion spectrum of this $Sp(k)$ gauge theory has a perturbative mixed gauge anomaly $-Z_4^{Spin(32)/\mathbb{Z}_2} q_1(G)$, which is canceled by the Green-Schwarz coupling $\int_{6d} B \wedge q_1(G)$. It is not immediate that there is no remaining uncanceled global mixed gauge anomaly, however.

In this paper we take the position that the consistency of string theory guarantees that the $Sp(k)$ gauge group is anomaly free, for both the perturbative and the global parts. In particular, we will assume that the anomaly polynomial of the gauged theory, where the $Sp(k)$ gauge fields are already path-integrated over, is simply obtained by dropping the $q_1(G)$-dependent terms from the total anomaly polynomial given above. What we will need in the next section is this anomaly polynomial of the gauged theory. We hope to come back to the issue of the global mixed gauge anomaly in the future.

## 2.3  NS5-branes in the $E_8 \times E_8$ heterotic string

A stack of $k$ coincident NS5-branes gives the rank-$k$ E-string theory, whose existence was first recognized in [GH96, SW96]. Its anomaly polynomial $I_8^{\text{rank-}k}$ was determined later in [OST14]. Taking the convention that the E-string theory corresponds to the instanton of the first $E_8$ factor whose instanton number is $n$ rather than $n'$, we find that

$$I_8^{\text{rank-}k} - kY_8^{E_8 \times E_8} = Z_4^{E_8 \times E_8} W_4^{E_8 \times E_8} \tag{2.14}$$

where

$$Z_4^{E_8 \times E_8} = X_4^{E_8 \times E_8} + k\chi(N), \qquad W_4^{E_8 \times E_8} = k\frac{p_1(T) + p_1(N)}{24} + \frac{\chi(N) - 2n + n'}{6}. \tag{2.15}$$

The $H$-field now satisfies

$$dH = Z_4^{E_8 \times E_8} = \frac{p_1(Q)}{2} - n - n' + k\chi(N) \tag{2.16}$$

where the explanation of the additional term $k\chi(N)$ is as before. We also learn that on the E-string worldvolume in the heterotic $E_8 \times E_8$ theory has an additional Green-Schwarz coupling

$$-\int_{6d} B \wedge W_4^{E_8 \times E_8}. \tag{2.17}$$

Then the combination of (2.16) and (2.17) produces an anomaly which cancels the difference (2.14) between the anomaly of the rank-$k$ E-string theory $I_8^{\text{rank-}k}$ and the inflow term $kY_8^{E_8 \times E_8}$.

## 2.4  Difference of the two cases

For simplicity, we only consider the case $k = 1$ and only keep the six-dimensional tangent bundle and the $SU(2)_R$ bundle to be nontrivial. We set the $SU(2)_L$ bundle and the $Spin(32)/\mathbb{Z}_2$ and $E_8 \times E_8$ gauge bundles to be trivial.

The ten-dimensional terms $X_4$ and $Y_8$ are now common to both $Spin(32)/\mathbb{Z}_2$ and $E_8 \times E_8$ theories,

$$X_4 = \frac{p_1(Q)}{2}, \qquad Y_8 = \frac{3p_1(Q)^2 - 4p_2(Q)}{192}. \tag{2.18}$$

The six-dimensional terms $Z_4^{Spin(32)/\mathbb{Z}_2}$ and $Z_4^{E_8 \times E_8}$ also become the same:

$$Z_4 = \frac{p_1(T)}{2} - c_2(R) \tag{2.19}$$

which in fact agrees also with $X_4$ above. The way the anomalies of the NS5-branes are reproduced is still different, since we have

$$I_8^{Sp(1)} - Y_8 = Z_4 \times \frac{p_1(T) - p_1(N)}{24}, \tag{2.20}$$

$$I_8^{\text{rank-1 E-string}} - Y_8 = Z_4 \times \left( \frac{p_1(T) + p_1(N)}{24} - \frac{c_2(R)}{6} \right). \tag{2.21}$$

Using $p_1(N) = -2c_2(R)$ under our simplifying assumptions, we have the difference

$$I_8^{sp(1)} - I_8^{\text{rank-1 E-string}} = Z_4 \times \frac{1}{3} c_2(R) \tag{2.22}$$

with the corresponding Green-Schwarz coupling

$$-\int_{6d} B \wedge \frac{1}{3} c_2(R) \tag{2.23}$$

for $dH = Z_4$. Our next aim is to convert this information into the determination of the global anomaly of the $SU(2)_R$ symmetry captured by the ten-dimensional $Sp(2)$ manifold with an $H$ flux, as discussed in the Introduction.

# 3 Evaluation of the global anomaly

In six dimensions, theories with $SU(2)$ symmetry can have global anomalies associated to

$$\pi_6(SU(2)) = \mathbb{Z}_{12}, \tag{3.1}$$

as was pointed out long time ago [Tos89, BV97]. A modern understanding of this global anomaly was provided in [LT20, DL20c]. The essential point is that there is *no* global anomaly if the spacetime is equipped only with the spin structure and the $SU(2)$ gauge field, since

$$\Omega_7^{\text{spin}}(BSU(2)) = 0. \tag{3.2}$$

This means that, before the introduction of the $H$ field, the anomaly is completely specified by the anomaly polynomial. The global anomaly only appears because we introduce the $H$ field satisfying $dH = Z_4$, where $Z_4$ is a degree-4 characteristic class constructed from the curvatures

of the spacetime and the $SU(2)$ gauge bundle. And the global anomaly of the theory with the $H$ field can be determined from the anomaly polynomial of the theory before the introduction of the $H$ field.[9]

We will use this formulation to compute the global anomaly associated to the difference of the anomaly polynomials (2.22). We will use the simplified notation $p_1 := p_1(T)$ and $c_2 := c_2(R)$ below.

## 3.1 Strategy

Let us consider the evaluation of the global anomaly of an $SU(2)$-symmetric theory whose anomaly polynomial is

$$I_8 = Z_4 W_4, \tag{3.3}$$

where $Z_4$ and $W_4$ are linear combinations of $p_1/2$ and $c_2$. We introduce a 3-form field

$$dH = Z_4 \tag{3.4}$$

and the Green-Schwarz coupling

$$-\int B \wedge W_4. \tag{3.5}$$

The inclusion of this Green-Schwarz term cancels the perturbative part of the anomaly, but the global part can remain. We are interested in determining it.

The total anomaly of this system is captured by a seven-dimensional invertible phase whose partition function is given by

$$X := \eta[M_7, Z_4 W_4] - \int_{M_7} H_3 W_4 \in \mathbb{R}/\mathbb{Z}. \tag{3.6}$$

Here, the first term, $\eta[M_7, I_8]$, is the eta invariant on the manifold $M_7$ (with the spin structure, gauge fields etc. implicitly specified) with the anomaly polynomial $I_8$, and the second term is the anomaly produced by the Green-Schwarz coupling. We evaluate this combination following [LT20, Sec. 3], which was based on [ST05, Appendix B], which was based on [BV97], which was based on various works in the late 80s, most notably [Tos89] and [ZOT88], which were ultimately based on [EN84]. As the previous descriptions were scattered throughout multiple references, here we try to combine them into one single narrative which is hopefully easier to follow.

For our purpose

$$Z_4 = \frac{p_1}{2} - 2c_2, \quad W_4 = \frac{1}{3}c_2 \tag{3.7}$$

as we saw above, while the computation performed in [LT20] was for

$$Z_4 = c_2, \qquad W_4 = \frac{1}{24}(\frac{p_1}{2} + c_2) \tag{3.8}$$

---

[9]This is perfectly analogous to the case of the global $\mathbb{Z}_2$ anomaly of $SU(2)$ in four dimensions, which can be determined from the anomaly polynomial of the $SU(3)$ theory [EN84] or the $U(2)$ theory [DL20a, DL20b].

which was for the anomaly of a Weyl fermion in the doublet of $SU(2)$, with a pseudo-Majorana condition imposed. For the moment let us work with an arbitrary choice of $Z_4$ and $W_4$. The steps are as follows.

1. We first define a homomorphism $X : \pi_6(SU(2)) \to \mathbb{R}/\mathbb{Z}$:

   (a) We restrict the manifold to be $M = S^7$.

   (b) We note that the combination $X$ (3.6) is independent of the choice of $H$ when we solve (3.4). Indeed, two solutions $H$ and $H'$ are related by $H = H' + dB$ for a *globally well-defined* $B$, since $H^3(S^7, \mathbb{R}) = 0$. Then the change in $X$ is $\int_M (dB) W_4 = \int_M B dW_4 = 0$. Therefore, $X$ is a functional of the manifold and the gauge field only. Note that this step fails for more complicated seven manifolds, for which the choice of $H$ field is not topologically unique.

   (c) $X$ has zero infinitesimal variation of the metric and of the gauge field, as the infinitesimal variation of the first term and that of the second term are by construction the same. Therefore, $X$ defines a map

$$X : \pi_6(SU(2)) \to \mathbb{R}/\mathbb{Z}. \tag{3.9}$$

   (d) $X$ is a homomorphism. To see this, denote by $(S^7, a)$ the $SU(2)$ configuration on $S^7$ specified by $a \in \pi_6(SU(2))$. Gluing $(S^7, a)$ and $(S^7, a')$ gives $(S^7, a + a')$. $X$ behaves additively under this operation too, since the $\eta$ invariant is additive and the $\int HW_4$ term is also additive.

2. We now lift $X : \pi_6(SU(2)) \to \mathbb{R}/\mathbb{Z}$ to $\tilde{X} : \pi_7(Sp(2)/SU(2)) \to \mathbb{R}$:

   (a) We utilize the fibration $SU(2) \to Sp(2) \to Sp(2)/SU(2) = S^7$ and the homotopy exact sequence associated to it given by

$$\cdots \to \underbrace{\pi_7(SU(2))}_{\mathbb{Z}_2} \xrightarrow{0} \underbrace{\pi_7(Sp(2))}_{\mathbb{Z}}$$

$$\xrightarrow[12\times]{\iota} \underbrace{\pi_7(Sp(2)/SU(2))}_{\mathbb{Z}} \xrightarrow[\text{mod } 12]{\partial}$$

$$\underbrace{\pi_6(SU(2))}_{\mathbb{Z}_{12}} \xrightarrow{0} \underbrace{\pi_6(Sp(2))}_{0} \to \cdots \tag{3.10}$$

   (b) We regard our $S^7$ to be on a boundary of $D^8$, so that $S^7 = \partial D^8$. It is not possible to extend the $SU(2)$ configuration specified by $a \in \pi_6(SU(2))$ to $D^8$, but it is possible to do so if we regard it as an $Sp(2) \supset SU(2)$ configuration. Such an extension is characterized by an element $\tilde{a} \in \pi_7(Sp(2)/SU(2)) = \mathbb{Z}$, such that $\partial(\tilde{a}) = a$.

(c) Let us now pick a set of $Sp(2)$-symmetric 6d fermions whose anomaly polynomial

$$\tilde{I}_8 = \frac{t}{48} \operatorname{tr} F^4 + J_8(c_2, p_1) \tag{3.11}$$

reduces to $P_8 = Z_4 W_4$ upon reducing to $Sp(1)$, by using

$$\operatorname{tr} F^4 = \frac{1}{2}(\operatorname{tr} F^2)^2 = 2(c_2)^2. \tag{3.12}$$

Here we define $c_2 = \operatorname{tr} F^2/2$ for both $Sp(2)$ and $Sp(1)$, and $t$ is a number.

(d) Using the Atiyah-Patodi-Singer index theorem, we find

$$X(a) = \tilde{X}(\tilde{a}) \pmod{\mathbb{Z}}, \tag{3.13}$$

where

$$X(a) = \eta[S^7, Z_4 W_4] - \int_{S^7} H W_4 \in \mathbb{R}/\mathbb{Z} \tag{3.14}$$

and

$$\tilde{X}(\tilde{a}) = \int_{D^8} (\tilde{I}_8 - Z_4 W_4) \in \mathbb{R}. \tag{3.15}$$

Note that $\tilde{X}(\tilde{a})$ is a homomorphism

$$\tilde{X} : \pi_7(Sp(2)/SU(2)) \to \mathbb{R}. \tag{3.16}$$

3. We will now evaluate $\tilde{X} : \pi_7(Sp(2)/SU(2)) \to \mathbb{R}$:

(a) As $\tilde{X}(\tilde{a})$ is a homomorphism to $\mathbb{R}$, it suffices to evaluate $\tilde{a} \in \pi_7(Sp(2)/SU(2))$ in the image of $\iota$ from $\pi_7(Sp(2))$. In particular, we have

$$\tilde{X}(\iota(1)) = \tilde{X}(12) = 12\tilde{X}(1) \in \mathbb{R} \tag{3.17}$$

and therefore

$$X(1) = \frac{1}{12}\tilde{X}(\iota(1)) \in \mathbb{R}/\mathbb{Z}. \tag{3.18}$$

(b) So let us calculate $\tilde{X}(\iota(1))$. Note that $\partial(\iota(1)) = 0$, meaning that the $SU(2)$ configuration on $S^7$ is trivial. Therefore it can also be extended trivially as $SU(2)$ configuration to $D^8$. We now form

$$\begin{aligned} S^8 = \ &D^8(\text{with } Sp(2) \text{ configuration specified by } \iota(1)) \\ &\sqcup D^8(\text{with } SU(2) \text{ configuration trivially extended}) \end{aligned} \tag{3.19}$$

(c) We now integrate $\tilde{I}_8 - Z_4 W_4$ on both sides of (3.19). As $\tilde{I}_8 - Z_4 W_4$ restricted to an $SU(2)$ configuration is zero, we have

$$\tilde{X}(\iota(1)) = \int_{S^8} (\tilde{I}_8 - Z_4 W_4). \tag{3.20}$$

(d) As $H^4(S^8, \mathbb{R}) = 0$, every degree-4 closed form on $S^8$ is cohomologically zero, and so the right hand side of the equation above drastically simplifies to give

$$\tilde{X}(\iota(\alpha)) = \int_{S^8} \frac{t}{48} \operatorname{tr} F^4 = t \tag{3.21}$$

where we used

$$\int_{S^8} \frac{1}{48} \operatorname{tr} F^4 = 1 \tag{3.22}$$

for the configuration $1 \in \pi_7(Sp(2)) \simeq \mathbb{Z}$.[10]

(e) Recalling (3.18), we find

$$X(1) = \frac{t}{12}. \tag{3.23}$$

## 3.2 Evaluation

To complete our analysis, we need to find an $Sp(2)$ fermion system whose anomaly becomes our

$$I_8^{\mathrm{diff}} = (\frac{p_1}{2} - 2c_2)\frac{1}{3}c_2 \tag{3.24}$$

when restricted to its $SU(2)$ subgroup, and compute the coefficient of $\operatorname{tr} F^4$. For this, we first express it as a linear combination of the anomaly polynomials of Weyl fermions in $\mathbf{1}$, $\mathbf{2}$ and $\mathbf{3}$ of $SU(2)$, which are given respectively by

$$I_{\mathbf{1}} = \frac{7p_1^2 - 4p_2}{5760}, \qquad I_{\mathbf{2}} = \frac{1}{24}p_1 c_2 + \frac{1}{12}c_2^2 + 2I_{\mathbf{1}}, \qquad I_{\mathbf{3}} = \frac{1}{6}p_1 c_2 + \frac{4}{3}c_2^2 + 3I_{\mathbf{1}}. \tag{3.25}$$

We find

$$I_8^{\mathrm{diff}} = -8I_{\mathbf{2}} + I_{\mathbf{3}} + 13I_{\mathbf{1}}. \tag{3.26}$$

Now, the branching rules from $Sp(2)$ to $SU(2)$ are

$$\mathbf{4} \to \mathbf{2} + 2 \cdot \mathbf{1}, \quad \mathbf{10} \to \mathbf{3} + 2 \cdot \mathbf{2} + 3 \cdot \mathbf{1}, \tag{3.27}$$

where $\mathbf{4}$ and $\mathbf{10}$ are the fundamental and adjoint representation of $Sp(2)$, respectively, see e.g. [Yam15]. From this we find

$$-10 \cdot \mathbf{4} + \mathbf{10} + 30 \cdot \mathbf{1} \longrightarrow -8 \cdot \mathbf{2} + \mathbf{3} + 13 \cdot \mathbf{1}. \tag{3.28}$$

Fermions in $\mathbf{4}$ and $\mathbf{10}$ of $Sp(2)$ have the following anomaly polynomials

$$I_{\mathbf{4}} = \frac{p_1}{48}c_2 + \frac{1}{24} \operatorname{tr} F^4 + 4I_{\mathbf{1}}, \qquad I_{\mathbf{10}} = \frac{p_1}{8}c_2 + \frac{1}{4} \operatorname{tr} F^4 + \frac{1}{16}(c_2)^2 + 10I_{\mathbf{1}} \tag{3.29}$$

---

[10] The normalization (3.22) follows from the following consideration. The $Sp(2)$ gauge configuration on $S^8$ determines an element of $\mathrm{KSp}(S^8)$, whose topological index is given by $\pi_8(BSp) = \pi_7(Sp)$. This equals, via the Atiyah-Singer index theorem, the analytic index defined using the number of zero modes of a fermion in the fundamental representation coupled to the gauge configuration. The expression of the analytic index as an integral of a differential form then yields the left hand side of (3.22).

from which we conclude that

$$\tilde{I}_8^{\text{diff}} = -10I_4 + I_{10} + 30I_1 = -\frac{1}{6}\operatorname{tr} F^4 + \cdots. \tag{3.30}$$

This means the number $t$ in (3.11) is $-8$, and we find that the global anomaly phase $\alpha$ associated to the global anomaly for $\pi_6(SU(2)) = \mathbb{Z}_{12}$ is

$$\alpha = \exp\left(-\frac{2\pi i}{12} \cdot 8\right) = \exp\left(\frac{2\pi i}{3}\right). \tag{3.31}$$

This is indeed nontrivial of order 3. This also shows, as we discussed in the Introduction, that the topological couplings of the gravitational sector of two supersymmetric heterotic string theories in ten dimensions differ by the $\mathbb{Z}_3$-valued discrete topological term detecting the same $Sp(2)$ with unit $H$ flux.

# 4 An alternative computation

Here we explain an alternative method of the determination of the discrete topological term, using the deformation of the NS5-branes into instantons of the non-Abelian gauge fields of heterotic string theories. This will allow us to determine that the non-tachyonic $Spin(16) \times Spin(16)$ heterotic string theory also has the nonzero discrete $\mathbb{Z}_3$-valued topological term.

## 4.1 NS5-branes as instanton configurations

So far we only considered the gravitational sector of heterotic string theories. There, the $Sp(2)$ group manifold with the unit $H$ flux is a generator of a $\mathbb{Z}_3$ subgroup of the string bordism group $\Omega_{10}^{\text{string}}$, and is not a boundary of any eleven-dimensional spin manifold with the $H$ field satisfying $dH = p_1(Q)/2$. The $\mathbb{Z}_3$-valued discrete topological term was associated to this configuration.

With an $SU(2)$ gauge field $F$, however, it is possible to find a smooth eleven-dimensional spin manifold with $H$ satisfying

$$dH = \frac{p_1(Q)}{2} + c_2(F) \tag{4.1}$$

instead. To see this, recall that a smooth one-instanton configuration in the heterotic string theory is a deformation of an NS5-brane, and the instanton charge is identified with the NS5-charge. Then, we can have a four-dimensional ball $B^4$ with a finite-sized one-instanton configuration at the center, such that we have $\int_{S^3} H = 1$ on the surrounding three-sphere. We can now fiber the entire setup over $S^7$, so that we have

$$B^4 \to N_{11} \to S^7 \tag{4.2}$$

whose boundary is

$$S^3 \to Sp(2) \to S^7, \tag{4.3}$$

effectively wrapping an instantonic NS5-brane on $S^7$ at the center of the $B^4$ fiber.

This means that the phase $\alpha$ associated by the $\mathbb{Z}_3$-valued topological term can alternatively computed using this smooth configuration. The method was explained in the physicists-oriented section of [TY23]. Let us briefly review this technique.[11]

## 4.2 Evaluation using instanton configurations: general theory

Suppose we have a set of $d$-dimensional fermions $\Psi$ charged under some symmetry $G$, with an anomaly described by an invertible theory $I$ in $(d+1)$ dimensions. This means that the partition function $Z_\Psi(M_d)$ of $\Psi$ on $M_d$ is a vector in the one-dimensional Hilbert space $\mathcal{H}_I(M_d)$ of the theory $I$ on $M_d$. Let us say that the introduction of the $H$ field with $dH = X_4$ trivializes this anomaly theory. This means that, given the data of the $H$ field on $M_d$, we have a canonical choice of the vector $v(M_d, H) \in \mathcal{H}_I(M_d)$. These vectors satisfy the consistency condition that

$$v(M_d, H) = \exp(2\pi i \int_{N_{d+1}} HY_{d-2}) Z_I(N_{d+1}) v(M'_d, H') \tag{4.4}$$

where $N_{d+1} : M'_d \to M_d$ is a bordism from the incoming boundary $M'_d$ to an outgoing boundary $M_d$, $Z_I(N_{d+1} : \mathcal{H}_I(M'_d) \to \mathcal{H}_I(M_d)$ is the associated evolution operator of the invertible theory, and $I_{d+2} = X_4 Y_{d-2}$ is the factorization of the anomaly polynomial.

With these vectors we can convert the fermion partition function into a complex number

$$\langle v(M_d, H), Z_\Psi(M_d) \rangle, \tag{4.5}$$

which reproduces the standard Green-Schwarz coupling in the following way. Let us say that we choose two different $H$ fields on the same $M_d$, given by $H_1 = H_0 + dB$, with $B$ a globally defined 2-form on $M_d$. We now take $N_{d+1} = M_d \times [0,1]$ and define $H$ on $N_{d+1}$ to be $H_0 + ds \wedge B$, where $s$ is the coordinate of the segment $[0,1]$. Then the condition (4.4) says

$$\langle v(M_d, H_1), Z_\Psi(M_d) \rangle = \exp(-2\pi i \int_{M_d} BY_{d-2}) \langle v(M_d, H_0), Z_\Psi(M_d) \rangle. \tag{4.6}$$

This is indeed the variation we expect.

We can also split (4.5) into the fermion contribution and the contribution from the Green-Schwarz coupling in favorable cases. Let us say $M_d$ is the boundary of a spin manifold $W_{d+1}$

---

[11]We note that the method is quite general and can be used to define a topological term for a spacetime structure $S$ when an anomaly $I$ for another spacetime structure $S'$ is given, assuming that the structure $S$ is an extension of $S'$, and that the anomaly $I$ trivializes when the structure $S'$ is extended to the structure $S$. This method was first described in [WWW17] when spacetime structures $S$, $S'$ consist of a common spacetime structure $\underline{S}$ and finite group symmetries $G$, $G'$ with a surjection $G \to G'$, assuming that the anomaly is described by ordinary cohomology. It was then extended to the case when the anomaly is given more generally by Anderson dual of bordism groups in [KOT19]. In the two references above, the motivation was to construct gapped boundary theories, but the methods work equally well when the extension from $S'$ to $S$ involves continuous fields. For example, the Wess-Zumino-Witten term is an example when the structure $S'$ is given by the spin structure with a continuous symmetry $G$, and the structure $S$ is given by adding a scalar field valued in the coset $G/H$; the application of this method yields a definition of the Wess-Zumino-Witten terms including the global topological part [Yon20]. The method explained below is when $S'$ consists of the spin structure and a continuous symmetry $G$ and $S$ extends this by adding a $B$-field.

without $G$ gauge field. In this case we have the Hartle-Hawking wavefunction of the invertible theory $Z_I(W_{d+1}) \in \mathcal{H}_I(M_d)$. Then the combination (4.5) can be split into

$$\langle v(M_d, H), Z_\Psi(M_d) \rangle = \langle v(M_d, H), Z_I(W_{d+1}) \rangle \langle Z_I(W_{d+1}), Z_\Psi(M_d) \rangle, \tag{4.7}$$

where the first factor is the phase produced by the Green-Schwarz coupling and the second factor is the fermion partition function. In this form, the anomaly of the individual factors is captured by the phase dependence of $Z_I(W_{d+1})$ on the choice of $W_{d+1}$.

The Green-Schwarz phase can be made more concrete when $M_d$ is also a boundary of $N_{d+1}$ equipped with the gauge field and the $H$ field solving $dH = X_4$. In this case, applying the consistency condition (4.4) for $M_d' = \varnothing$, we find

$$\langle v(M_d, H), Z_I(W_{d+1}) \rangle = \exp(-2\pi i \int_{N_{d+1}} HY_{d-2}) \langle Z_I(N_{d+1}), Z_I(W_{d+1}) \rangle \tag{4.8}$$

$$= \exp\left(-2\pi i \int_{N_{d+1}} HY_{d-2} + 2\pi i \eta_I[\overline{N_{d+1}} \sqcup W_{d+1}]\right), \tag{4.9}$$

where

$$\exp(2\pi i \eta[M_{d+1}]) = Z_I(M_{d+1}) \tag{4.10}$$

is the partition function of the invertible theory on the closed manifold $M_{d+1}$ equipped with the spin structure, the $G$ gauge field etc., which are typically given by an eta invariant.

## 4.3 Evaluation using instanton configurations: our case

To apply this general technique in our case, we need a few more preparations. Ten-dimensional anomaly polynomials were reviewed in Sec. 2.1. We now restrict the gauge bundles to the common $\mathfrak{so}(16) \times \mathfrak{so}(16)$ subalgebra. Denote by $p_{1,2}$ and $p'_{1,2}$ the Pontryagin classes of two $so(16)$ bundles. Then we have $n = p_1/2$, $n' = p'_1/2$ for the instanton numbers of the $E_8 \times E_8$ theory, and $p_1(F) = p_1 + p'_1$ and $p_2(F) = p_2 + p_1 p'_1 + p'_2$ for the $Spin(32)/\mathbb{Z}_2$ theory. Under this reduction, we have

$$X_4 := X_4^{Spin(32)/\mathbb{Z}_2} = X_4^{E_8 \times E_8} = \frac{p_1(Q)}{2} - n - n'. \tag{4.11}$$

We are interested in the difference of the Green-Schwarz contributions, so we consider

$$I_{12}^{\text{diff}} := I_{12}^{Spin(32)/\mathbb{Z}_2} - I_{12}^{E_8 \times E_8} = X_4 Y_8^{\text{diff}} \tag{4.12}$$

where

$$Y_8^{\text{diff}} = Y_8^{Spin(32)/\mathbb{Z}_2} - Y_8^{E_8 \times E_8} = \frac{n^2 + nn' + n'^2 - p_2 - p'_2}{6}. \tag{4.13}$$

For our purposes, we only need to turn on the gauge configuration in an $su(2) \subset su(2) \times su(2) \simeq so(4) \subset so(16)$ subalgebra. Then we can set $n = -c_2(F)$, $n' = 0$ and $p_2 = p'_2 = 0$, further simplifying the expression above so that

$$X_4 = \frac{p_1(Q)}{2} + c_2(F), \qquad Y_8^{\text{diff}} = \frac{c_2(F)^2}{6}. \tag{4.14}$$

We now use $M_{10} = Sp(2)$ equipped with the standard choice of the $H$ field, with $N_{11}$ constructed in (4.2), equipped with the instanton configuration as described above. For $W_{11}$, we take the same manifold $N_{11}$ but without the instanton configuration and the $H$ field. Then the expression (4.9) should give the same phase $\alpha$ as computed in the previous section.[12]

## 4.4  Discrete topological term of the $Spin(16) \times Spin(16)$ theory

This alternative method allows us to determine that the non-tachyonic $Spin(16) \times Spin(16)$ heterotic string theory has the $\mathbb{Z}_3$-valued discrete topological term. This is based on the fact[13] that the massless fermion spectrum of the non-tachyonic $Spin(16) \times Spin(16)$ theory is given by that of the $Spin(32)/\mathbb{Z}_2$ supersymmetric heterotic string together with that of the chirality flipped version of the $E_8 \times E_8$ supersymmetric heterotic string, both restricted to the common $Spin(16) \times Spin(16)$ part. This in particular means that the anomaly polynomial $I_{12}^{Spin(16) \times Spin(16)}$ of the non-tachyonic $Spin(16) \times Spin(16)$ theory is the difference $I_{12}^{\text{diff}}$, given in (4.12), of the anomaly polynomials of the $Spin(32)/\mathbb{Z}_2$ and $E_8 \times E_8$ supersymmetric heterotic strings. In contrast, in the other approach which directly uses the anomaly on the NS5 brane worldvolume theory, the meaning of taking such a formal difference between two heterotic string theories is more opaque. This means that it is harder to argue for a direct connection to the NS5 brane in the $Spin(16) \times Spin(16)$ heterotic string theory. [14]

Then, this means that the $\mathbb{Z}_3$-valued discrete topological term of the $Spin(16) \times Spin(16)$ theory is given by the formula (4.9) with the same $I_{12}^{\text{diff}}$ and the same eta invariant used in the case of the discrete difference of the topological terms of the two supersymmetric heterotic string theories. It therefore has the same value, namely $\exp(2\pi i/3)$.

# 5  Relation to topological modular forms

Finally let us explain the relation of our findings here to the theory of topological modular forms. Topological modular forms (TMFs) are objects in algebraic topology, developed partially in order to explain the topological origin of the behavior of the elliptic genus of manifolds noticed in [Wit87], and partially following a trend internal to algebraic topology. The proposal of Stolz and Teichner [ST04, ST11] says that the deformation classes $[T]$ of two-dimensional spin-modular-invariant $\mathcal{N}=(0,1)$ supersymmetric theories $T$ with gravitational anomaly $n\frac{p_1}{48}$ form the group $\mathrm{TMF}_n$ of topological modular forms of degree $n$. Here the gravitational anomaly is normalized

---

[12]In fact, it was this method using (4.9) that occurred first on the minds of the authors in order to evaluate the $\mathbb{Z}_3$-valued discrete topological term. Although all the ingredients in (4.9) are mathematically well-defined, the evaluation of the two contributions in (4.9) turned out to be beyond the capabilities of the authors, at least at present. The indirect method using the known anomaly polynomials of the NS5-branes in the two supersymmetric heterotic string theories was the second approach the authors took. The authors found it interesting that the duality of various string theories can be used to circumvent a difficult, direct evaluation of this mathematical expression.

[13]There is a way to understand this fact using a $\mathbb{Z}_2$ gauging on the worldsheet, see [BSLTZ23].

[14]See also [BL23] for discussions on Dai-Freed anomalies in compactifications of $Spin(16) \times Spin(16)$ heterotic string theory.

so that we have $n = 2(c_R - c_L)$ when the theory is superconformal. Various pieces of evidence supporting this proposal have accumulated in the last several years, see e.g. [GPPV18, GJFW19, GJF19, Tac21, LP21, Yon22, AKL22].

In particular, when $n$ is even, there is a map

$$\Phi \colon \mathrm{TMF}_n \to \mathrm{MF}_{n/2}, \tag{5.1}$$

where $\mathrm{MF}_d$ is the set of modular forms of degree $d$. This map extracts the elliptic genus of the theory in the following sense under the Stolz-Teichner proposal. Namely, we have

$$\Phi([T]) = \eta(q)^n Z_{\mathrm{ell}}^T(q) \tag{5.2}$$

where $Z_{\mathrm{ell}}^T(q)$ is the elliptic genus of the theory $T$ as usually defined by physicists.

For $n$ even, let us define

$$A_n = \mathrm{Ker}(\Phi \colon \mathrm{TMF}_n \to \mathrm{MF}_{n/2}). \tag{5.3}$$

$A_n$ captures the information about subtle $\mathcal{N}{=}(0,1)$ theories which are nontrivial even though their elliptic genera are trivial. $A_n$ is known to be a finite Abelian group.

The internal worldsheet theories of two ten-dimensional supersymmetric heterotic strings have $(c_L, c_R) = (16, 0)$ and therefore define elements of $\mathrm{TMF}_{-32}$. Denote these two classes as $[T^{E_8 \times E_8}]$ and $[T^{Spin(32)/\mathbb{Z}_2}]$. What are the relation between these two TMF elements? As the two theories $T^{E_8 \times E_8}$ and $T^{Spin(32)/\mathbb{Z}_2}$ have the same partition function and therefore the elliptic genus, we have

$$T^{Spin(32)/\mathbb{Z}_2} - T^{E_8 \times E_8} \in A_{-32}. \tag{5.4}$$

The detailed computations of $\mathrm{TMF}_n$ by mathematicians shows that

$$A_{-32} = \mathbb{Z}_3, \tag{5.5}$$

see e.g. [BR21]. Our main result amounts to the demonstration that $[T^{Spin(32)/\mathbb{Z}_2}] - [T^{E_8 \times E_8}]$ is the generator of this $\mathbb{Z}_3$.

This is because of the following. Mathematicians knew [BR21] that the Anderson self-duality of topological modular forms leads, among other things, to the statement that the two finite groups $A_{-d-22}$ and $A_d$ are Pontryagin dual to each other when $d$ is even. Equivalently, there is a perfect pairing

$$((x, y)) \in U(1) \tag{5.6}$$

for $x \in A_{-d-22}$ and $y \in A_d$. In [TY23], it was shown that this perfect pairing gives the exponentiated Euclidean action for the discrete Green-Schwarz effect when $x = [T]$ is the internal worldsheet theory with zero elliptic genus for the heterotic compactification down to $d$ dimensions and $y = [M]$ is the worldsheet sigma model for the $d$-dimensional space $M$ equipped with an appropriate $H$ field which is null bordant as a spin manifold. Our computation in this paper shows that

$$(([T^{Spin(32)/\mathbb{Z}_2}] - [T^{E_8 \times E_8}], [Sp(2)])) = e^{\pm 2\pi i/3}, \tag{5.7}$$

showing simultaneously that $[T^{Spin(32)/\mathbb{Z}_2}] - [T^{E_8 \times E_8}]$ is a generator of $A_{-32} = \mathbb{Z}_3$ and $[Sp(2)]$ is a generator of $A_{10} = \mathbb{Z}_3$. Our argument also shows that the class $[T^{Spin(16) \times Spin(16)}]$ of the worldsheet theory of the non-tachyonic non-supersymmetric heterotic string theory satisfies

$$[T^{Spin(16) \times Spin(16)}] = [T^{Spin(32)/\mathbb{Z}_2}] - [T^{E_8 \times E_8}] \tag{5.8}$$

and is a generator of $A_{-32}$.

# Acknowledgments

YT and HYZ are supported in part by WPI Initiative, MEXT, Japan at Kavli IPMU, the University of Tokyo.

# A  Summary for mathematicians

## A.1  Mathematical-physical context

In this appendix we explain the content of the paper in a language hopefully more understandable to mathematicians. Let us first recall the relevant parts of the content of [TY23].

In that paper, the Anderson self-duality of topological modular forms is formulated as the statement

$$\mathrm{KO}((q))/\mathrm{TMF} \simeq \Sigma^{-20} I_{\mathbb{Z}} \mathrm{TMF}, \tag{A.1}$$

where the left hand side was defined by the cofiber sequence

$$\mathrm{TMF} \xrightarrow{\Phi} \mathrm{KO}((q)) \longrightarrow \mathrm{KO}((q))/\mathrm{TMF}. \tag{A.2}$$

Here $\Phi : \mathrm{TMF} \to \mathrm{KO}((q))$ is the standard morphism corresponding to taking the Tate curve (see [AHR10, HL13] for more details of this morphism). Let us now define

$$A_d := \mathrm{Ker}(\Phi : \pi_d \mathrm{TMF} \to \pi_d \mathrm{KO}((q))), \tag{A.3}$$

which is known to be a finite group. One consequence of the self-duality was that there is a perfect pairing

$$((-,-)) : A_{-22-d} \times A_d \to \mathbb{Q}/\mathbb{Z} \tag{A.4}$$

unless $d \equiv 3, -1 \bmod 24$. In the range $-32 \leq d \leq 10$, the groups $A_d$ are non-trivial in the following places:

| $d$ | 3 | 6 | 8 | 9 | 10 |
|---|---|---|---|---|---|
| $A_d$ | $\mathbb{Z}/24$ | $\mathbb{Z}/2$ | $\mathbb{Z}/2$ | $\mathbb{Z}/2$ | $\mathbb{Z}/3$ |
| $M_d$ | $SU(2)$ | $SU(2)^2$ | $SU(3)$ | $U(3)$ | $Sp(2)$ |
| $d$ | | $-28$ | $-30$ | $-31$ | $-32$ |
| $A_d$ | | $\mathbb{Z}/2$ | $\mathbb{Z}/2$ | $\mathbb{Z}/2$ | $\mathbb{Z}/3$ |

$$\tag{A.5}$$

There, for positive degrees, we also listed the Lie groups $M_d$, assumed to be equipped with the Lie group framing, such that $\sigma_{\text{string}}([M_d])$ is the generator of $A_d$, where

$$\sigma_{\text{string}} : \text{MString} \to \text{TMF} \tag{A.6}$$

is the sigma orientation of [AHR10]. It would then be interesting to have explicit generators in the negative degrees.

To give explicit classes in the negative degrees, we use the following conjecture, which is a special case of a broader proposal by Stolz and Teichner. A more precise formulation was given in [TY23], but here the following version would suffice:

**Conjecture A.7** *A self-dual vertex operator superalgebra $\mathbb{V}$ of central charge $c$ gives a class $[\mathbb{V}] \in \pi_{-2c}\text{TMF}$, whose image $\Phi([\mathbb{V}]) \in \pi_{-2c}\text{KO}((q))$ is fixed in terms of the character theory of $\mathbb{V}$.*

In particular, whether $[\mathbb{V}] \in A_{-2c}$ or not can be easily found from the knowledge of $\mathbb{V}$.

Now, self-dual vertex operator superalgebras $\mathbb{V}$ of $c \leq 24$ were recently classified in [HM23]. In the range $c \leq 16$, corresponding to $-32 \leq d < 0$ in terms of the TMF degree $d$, those $\mathbb{V}$ that i) are not a product of two self-dual vertex operator superalgebras of lower central charges and ii) $\Phi([\mathbb{V}]) \in A_d$ were found to exist only at

$$c = 1/2, 14, 15, 31/2, 16, \tag{A.8}$$

corresponding to

$$d = -1, -28, -30, -31, -32. \tag{A.9}$$

This miraculously matches the place where $A_{d<-1}$ is nontrivial in (A.5). This strongly suggests that the corresponding $[\mathbb{V}]$ for $d = -28, -30, -31$ and $-32$ are the generators of $A_d$. In [TY23], we provided evidence for the cases $d = -28, -30$ and $-31$. The present paper provides evidence for the case $d = -32$.

## A.2   Mathematical translation of the content of the paper

One way to check that $[\mathbb{V}]$ is a generator of $A_{-32} = \mathbb{Z}/3$ is to compute the pairing (A.4) with the generator $\sigma_{\text{string}}([Sp(2)])$ of $A_{10} = \mathbb{Z}/3$ and show it to be nontrivial. For this purpose we use the following theorem proved in [TY23]:

**Theorem A.10** *Suppose that $x \in \pi_{-d-22}\text{TMF} = \text{TMF}^{d+22}(pt)$ lifts to $\tilde{x} \in \text{TMF}^{d+22+k}(BG)$ and that the string manifold $M_d$ is such that it is null when sent to $\text{MString}_{d+k}(BG)$, where $k : BG \to K(\mathbb{Z}, 4)$ specifies the twist. Let $N_{d+1}$ be a $(BG, k)$-twisted string manifold whose boundary is $M_d$. Then there is an explicit differential geometric formula which computes the pairing*

$$((x, \sigma_{\text{string}}([M_d]))) \tag{A.11}$$

*such that it only depends on $\Phi(\tilde{x}) \in \text{KO}((q))^{d+22}(BG)$ and the $(BG, k)$-twisted string manifold $N_{d+1}$.*

In our convention, a $(X, k)$-twisted string manifold $M$ for $k \in H^4(X, \mathbb{Z})$ comes with a map $f : M \to X$ such that $p_1/2 - f^*(k)$ is trivialized, among other things.

To apply this theorem to $[\mathbb{V}] \in \pi_{-2c}\text{TMF}$ one needs an equivariant version of Conjecture A.7, which was also given in [TY23]:

**Conjecture A.12** *A self-dual vertex operator superalgebra $\mathbb{V}$ of central charge $c$, containing a simple affine Lie algebra $\hat{\mathfrak{g}}$ at level $k$, gives a class $[\mathbb{V}] \in \text{TMF}^{2c+k\tau}(BG)$, where $G$ is the simply-connected simple compact Lie group corresponding to $\mathfrak{g}$ and $\tau : BG \to K(\mathbb{Z}, 4)$ is such that its class $[\tau] \in H^4(BG, \mathbb{Z}) \simeq \mathbb{Z}$ is the generator. Furthermore, the image $\Phi([\mathbb{V}]) \in \text{KO}((q))^{2c}(BG)$ is fixed in terms of the character theory of $\mathbb{V}$.*

This theorem was applied in [TY23] in the case of the element $[\mathbb{V}]$ for $d = -28$ and the six-dimensional string manifold $M_6 = SU(2)^2$, using $G = SU(2)$, successfully showing that the pairing was nontrivial.

In our case of $c = 16$, the relevant $\mathbb{V}$ is a particular extension of $\widehat{\mathfrak{so}}(16)_1 \times \widehat{\mathfrak{so}}(16)_1$. For this reason we denote it as $\mathbb{V}_{D_8 \times D_8}$. This vertex operator superalgebra also contains $\widehat{\mathfrak{su}}(2)_1$, and as such, this vertex operator superalgebra defines an element $[\mathbb{V}_{D_8 \times D_8}] \in \text{TMF}^{32+\tau}(BSU(2))$. We can also construct a null-bordism $N_{11}$ of the string manifold $M_{10} = Sp(2)$ as a $(BSU(2), \tau)$-twisted string manifold. Then we can apply Theorem A.10 above. Unfortunately the resulting formula in this case, although explicit, was too complicated for the authors to evaluate.

Instead we use the following, more indirect approach. As is well-known, there are two even self-dual lattices at rank 16, of type $E_8 \times E_8$ and $D_{16}$. Correspondingly, there are two self-dual vertex operator algebras, which we denote as $\mathbb{V}_{E_8 \times E_8}$ and $\mathbb{V}_{D_{16}}$. They contain the affine Lie algebras $(\widehat{\mathfrak{e}}_8)_1 \times (\widehat{\mathfrak{e}}_8)_1$ and $\widehat{\mathfrak{so}}(32)_1$ respectively. Therefore, they both define an element of $\text{TMF}^{32+\tau}(BSU(2))$. Using their character theories, it is straightforward to check that

$$\Phi([\mathbb{V}_{D_{16}}]) - \Phi([\mathbb{V}_{E_8 \times E_8}]) = \Phi([\mathbb{V}_{D_8 \times D_8}]) \in \text{KO}((q))^{32}(BSU(2)). \tag{A.13}$$

From Theorem A.10, we then have

$$(([\mathbb{V}_{D_{16}}] - [\mathbb{V}_{E_8 \times E_8}], \sigma_{\text{string}}[Sp(2)])) = (([\mathbb{V}_{D_8 \times D_8}], \sigma_{\text{string}}[Sp(2)])). \tag{A.14}$$

We now use the left hand side to evaluate the right hand side.

For the left hand side, we use another proposition proved in [TY23]:

**Proposition A.15** *For $x \in \pi_{-d-22}\text{TMF}$ and a string manifold $M_d$ which is null as a spin manifold, we have*

$$((x, \sigma_{\text{string}}([M_d]))) = \langle \alpha_{\text{spin/string}}(x), [N_{d+1}, M_d] \rangle, \tag{A.16}$$

*where $\alpha_{\text{spin/string}} : \pi_{-d-22}\text{TMF} \to \pi_{-d-2}I_{\mathbb{Z}}\text{MSpin}/\text{MString}$ is a morphism constructed in [TY23], $N_{d+1}$ is the spin null bordism of $M_d$ such that $\partial N_{d+1} = M_d$, $[N_{d+1}, M_d]$ is the relative bordism class in $\pi_{d+1}\text{MSpin}/\text{MString}$, and*

$$\langle -, - \rangle : (\pi_{-d-2}I_{\mathbb{Z}}E)_{\text{torsion}} \times (\pi_{d+1}E)_{\text{torsion}} \to \mathbb{Q}/\mathbb{Z} \tag{A.17}$$

*is the torsion pairing induced by the Anderson duality.*

To evaluate the right hand side of (A.16), we use another trick. We know $\pi_4\mathrm{MSpin/MString} = (\mathrm{MSpin/MString})^{-4}(pt) = \mathbb{Z}$ is generated by $a := [D^4, S^3]$ where the string structure of the boundary $S^3$ is given by the Lie group framing. By considering the action of $SU(2)$ on $S^3$, this element can actually be lifted to an element

$$a \in (\mathrm{MSpin/MString})^{-4+2c_2}(BSU(2)), \tag{A.18}$$

for which we used the same symbol by a slight abuse of the notation. The slant product by $a$ then gives a homomorphism

$$a : \mathrm{MString}_{d-3+2c_2}(BSU(2)) \to (\mathrm{MSpin/MString})_{d+1}(pt), \tag{A.19}$$

together with its Anderson dual

$$I_{\mathbb{Z}}a : (I_{\mathbb{Z}}\mathrm{MSpin/MString})^{d+2}(pt) \to (I_{\mathbb{Z}}\mathrm{MString})^{d-2+2c_2}(BSU(2)). \tag{A.20}$$

Now, $Sp(2)$ as a string manifold has a fibration structure

$$S^3 \to Sp(2) \to S^7. \tag{A.21}$$

where $S^7$ has a natural $(BSU(2), 2c_2)$-twisted string structure, which we collectively denote as $(S^7, f)$. A part of the data contained in $f$ is the classifying map $f : S^7 \to BSU(2)$, which is known to be given by the generator of $\pi_7(BSU(2)) = \pi_6(SU(2)) = \mathbb{Z}/12$. Let us denote its class by $[S^7, f] \in \mathrm{MString}_{7+2c_2}(BSU(2))$. The discussions above amount to the statement

$$a([S^7, f]) = [N_{11}, Sp(2)]. \tag{A.22}$$

From the naturality of the Anderson dual pairing, we now have

$$\begin{aligned}\langle \alpha_{\mathrm{spin/string}}(x), [N_{d+1}, Sp(2)] \rangle &= \langle \alpha_{\mathrm{spin/string}}(x), a([S^7, f]) \rangle \\ &= \langle (I_{\mathbb{Z}}a \circ \alpha_{\mathrm{spin/string}})(x), [S^7, f] \rangle \end{aligned} \tag{A.23}$$

So our task is now reduced to the evaluation of

$$I_{\mathbb{Z}}a \circ \alpha_{\mathrm{spin/string}} : \mathrm{TMF}^{d+22}(pt) \to (I_{\mathbb{Z}}\mathrm{MString})^{d-2+2c_2}(BSU(2)) \tag{A.24}$$

applied to $x = [\mathbb{V}_{D_{16}}] - [\mathbb{V}_{E_8 \times E_8}] \in \mathrm{TMF}^{32}(pt)$.

Here comes the string-theoretical information which is hard to translate. It just so happens that string theorists explicitly know the elements

$$z_{D_{16}} = I_{\mathbb{Z}}a \circ \alpha_{\mathrm{spin/string}}([\mathbb{V}_{D_{16}}]), \qquad z_{E_8 \times E_8} = I_{\mathbb{Z}}a \circ \alpha_{\mathrm{spin/string}}([\mathbb{V}_{E_8 \times E_8}]). \tag{A.25}$$

Combining all the discussions so far, we have the equality

$$((\mathbb{V}_{D_8 \times D_8}], [Sp(2)])) = \langle z_{D_{16}} - z_{E_8 \times E_8}, [S^7, f] \rangle \tag{A.26}$$

whose right hand side can be evaluated with some efforts. The description of elements $z_{D_{16}}$, $z_{E_8 \times E_8}$ fills Sec 2, and the computation of the right hand side of (A.26) is the topic of Sec. 3. We

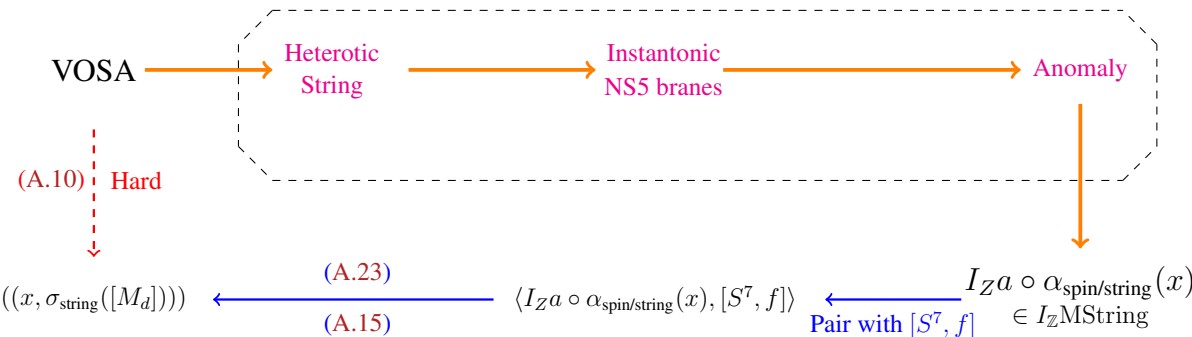

Figure 2: Summary of the main idea in this appendix. On the left, we have the hard problem of taking the VOSA and identifying a pairing as given in Theorem A.10. This would rigorously show that the VOSA represents a torsional element in $\pi_{-2c}\text{TMF}$ for $c = 16$, as explained in Conjecture A.7. However, the direct mathematical implementation is not tractable by the authors. Therefore, the author take the indirect, physical approach, where the steps that do not yet admit full mathematical formulations are written in magenta and labeled differently.

conclude that the pairing (A.26) is a nontrivial cubic root of unity, which was what we wanted to show.

Let us end this appendix by explaining how physicists know the two elements $z_{D_{16}}$, $z_{E_8 \times E_8}$. A ten-dimensional heterotic string theory is specified by a choice of the self-dual vertex operator superalgebra $\mathbb{V}$ of $c = 16$ to be used as the theory on the worldsheet. Such a string theory is known to possess a six-dimensional 'brane' called the NS5-brane, where the shift of the dimensionality between six and five by one is caused by the convention that a $p$-brane refers to an extended object with $p$ spatial and 1 temporal dimensions. The local geometry of the NS5-brane is given by a fibration of $\mathbb{R}^4 \setminus \{0\}$ over a six-dimensional space $M_6$, such that we have

$$\int_{S^3} H = 1 \tag{A.27}$$

on any $S^3$ surrounding the origin in the fiber. A six-dimensional quantum field theory is known to reside 'on the zero section' of this fibration, morally speaking.

The heterotic string theory we use $\mathbb{V}_{D_{16}}$ or $\mathbb{V}_{E_8 \times E_8}$ as the worldsheet theory is known as the $Spin(32)/\mathbb{Z}_2$ or $E_8 \times E_8$ heterotic string theory, respectively. The physics of the NS5-branes in these two cases is sufficiently well-understood, to the extent that we know their anomalies as an element of $(I_{\mathbb{Z}}\text{MString})^{8+2c_2}(BSU(2))$, These are the two elements $z_{D_{16}}$, $z_{E_8 \times E_8}$ we referred to above. The determination of these two elements use various string dualities which bring us outside of the realm of heterotic string theory. Namely, to determine $z_{D_{16}}$ the duality to Type I string was used, and to determine $z_{E_8 \times E_8}$ the duality to M theory was used. The main idea

in this appendix of using physical input to indirectly solve s difficult mathematical problem is schematically summarized in figure 2.

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
