# Peer review of "On a $\mathbb{Z}_3$-valued discrete topological term in 10d heterotic string theories"

_SciPost Physics_

## Round 2 · Referee Report · Anonymous (Referee 1) · 2024-5-24

# Report for *SciPost* on Tachikawa and Zhang, "On a $\mathbb{Z}_3$-valued discrete topological term in 10d heterotic string theories"

This is a nice paper, which deserves publication. Let me try to restate the main result. There are two important heterotic string theories, called $E_8 \times E_8$ and $Spin(32)/\mathbb{Z}_2$, defined as having as their worldsheet theories the bosonic holomorphic VOAs of the same names. These VOAs have the same character; and so much of the physics of the string theories match. What the authors argue is that the effective actions do differ: specifically, in the difference of their effective actions is a $\mathbb{Z}_3$-valued discrete topological term. The language of the paper is heavily string theoretic, but then the authors supply a section for a reader who comes originally from mathematics, explaining the structure of their results in terms of (a conjectural description of) TMF. This referee found that section clarifying.

The authors had originally hoped to directly compute this difference as follows. There are maps $Spin(16) \times Spin(16) \to E_8 \times E_8$ and $Spin(16) \times Spin(16) \to Spin(32)/\mathbb{Z}_2$ (with slightly different kernels). Take the effective gauge theory actions for the two theories, and restrict along these maps. One finds two different effective gauge theory actions for $Spin(16) \times Spin(16)$. One could hope to directly compute the difference of these actions. The authors, sadly, report that this direct computation eluded them. Rather, the authors input more string theoretic knowledge and offer a less direct computation, which compactifies to 6D (along a single NS5-brane) and looks at an $SU(2)$ rotating the normal bundle, where the topological term can be seen visible. The abstract and introduction could be interpreted as suggesting that the authors supply two independent arguments for their result, one from $SU(2)$ and one from $Spin(16) \times Spin(16)$. I recommend clarifying that the results about the non-tachyonic non-supersymmetric $Spin(16) \times Spin(16)$ heterotic string are arrived at as corollaries of the authors calculation, and not as independent verification.

There is one place to be cautious about the authors' results: as a critical point (equations 2.20–2.22), there is a vital sign, and if a sign error was made, then the result would fail. I did not find any sign error, and I do not suggest that there is one. But signs are notoriously difficult to track, so this point makes me slightly nervous.

The following is a list of small typos and other comments.

1. In footnote 2, I have trouble parsing "the topological parts of the effective action ... depend continuously on the metric ...".

2. On page 3, "shrinks to zero side" should be "shrinks to zero size".

3. On page 4, "the $H$-filed" should be "the $H$-field".

4. On page 5, "is concerned the anomaly" should be "is concerned with the anomaly". I also suggest "we do need" in place of "we only need".

5. On page 6, when you write "Then our main result can be summarized as saying", it is not at all obvious yet why this is true. Of course, the point here is that you are outlining the rest of the paper, and I should wait for later explanation. But as written, I could have hoped that maybe you were doing the full translation to TMF now. Maybe a reference to the appropriate section would be appropriate?

6. On page 24, the sentence starting "The heterotic string theory we use" doesn't quite parse. The following sentence ends with a comma, but should end with a period.

---

## Round 2 · Referee Report · Anonymous (Referee 2) · 2024-6-24

Report

The authors show that the 10d effective actions of $E_8\times E_8$ and $SO(32)$ heterotic string theories, with gauge fields turned off, have different $\mathbb{Z}_3$-valued topological terms. The difference can be detected by putting the theories on the spacetime which is the $Sp(2)$ group manifold with a non-trivial 3-form flux. To determine the difference, the authors relate 10d topological terms to the global anomalies of the NS5-brane worldvolume theories via Green-Schwarz mechanism. These global anomalies are in turn related to the perturbative anomalies (known in part from previous works) via an Elitzur-Nair-like relation.

The authors also comment on possible alternative approaches to determine the difference.

I believe that the results and the techniques used in the manuscript are quite interesting for theoretical physicists working on string theory, global anomalies, and related topics and possibly also for algebraic topologists. I find the manuscript well-written. I would like to recommend it for publication.

Requested changes

I have only very minor suggestions:

1) I think it would be better to add some brief explanation or references about the origin of the formulas (2.1)-(2.6).

2) The convention for ball/disk is not very consistent, e.g. on page 4 it is a disk $D^4$, while on page 15 it is a ball $B^4$.

3) In point 2(c) I think it would be better to mention that $t$ and $J_8$ will fixed later on. Otherwise, I found that part a bit confusing (until I reached the end of (3.2)).

4) After (5.1) the authors write "... set of modular forms of degree $d$." I think "weight" is a much more commonly used term in the context of usual modular forms (instead of "degree", which is more appropriate in the cohomology theory context) .

I've also noticed some misprints:

6) middle of page 5: I think "be" is missing in "... can only detected ..."

7) missing closing parenthesis in the second line after eq. (4.4)

8) sentence on page 25: "... solve s difficult ..."

Recommendation

Ask for minor revision

---

## Editorial Decision

resubmitted